

# Spatiotemporal variables comparison between drop jump and horizontal drop jump in elite jumpers and sprinters

Raynier Montoro-Bombú[1], Paulo Miranda-Oliveira[2,3,4], Maria João Valamatos[5,6], Filipa João[5,6], Tom JW Buurke[7,8], Amândio Cupido Santos[1] and Luis Rama[1]

[1] University of Coimbra, Research Unit for Sport and Physical Activity (CIDAF), Faculty of Sport Sciences and Physical Education, Coímbra, Portugal
[2] Portuguese Athletics Federation (FPA), Lisboa, Lisboa, Portugal
[3] School of Technology and Management (ESTG), Polytechnic of Leiria, Leiria, Portugal
[4] Interdisciplinary Research Centre Egas Moniz (CIIEM), Egas Moniz School of Health & Science, Almada, Portugal
[5] Centro Interdisciplinar para o Estudo da Performance Humana (CIPER), Faculdade de Motricidade Humana, Lisboa, Cruz-Quebrada, Portugal
[6] Laboratório de Biomecânica e Morfologia Funcional, Faculdade de Motricidade Humana, Universidade de Lisboa, Estrada da Costa, Lisboa, Cruz-Quebrada, Portugal
[7] University of Groningen, University Medical Center Groningen, Department of Human Movement Sciences, Groningen, Netherlands
[8] KU Leuven, Department of Movement Sciences, Leuven, Belgium

Corresponding author
Raynier Montoro-Bombú,
rayniermb@gmail.com

## ABSTRACT

**Background:** General expectations speculated that there are differences between drop jump (DJ) and horizontal drop jump (HDJ) exercises. While these criteria may be valid, we have yet to find a report that explores these differences in competitive level athletes.

**Objective:** The study aimed to compare spatiotemporal variables in the drop jump (DJ) *vs.* the horizontal drop jump (HDJ) in elite jumpers and sprinters.

**Methods:** Sixteen international-level male athletes performed two DJ attempts at different fall heights 0.3, 0.4, and 0.5 m (DJ30, DJ40, and DJ50), and after 2 h, they performed two HDJ attempts (HDJ30, HDJ40, HDJ50). All jumps were performed on a Kistler force plate. The variables analyzed were ground contact time (GCT), flight time (FT), eccentric phase time, concentric phase time, and time to peak concentric force.

**Results:** The GCT was found to be significantly shorter in DJ *vs.* HDJ ($Z = 4.980$; $p = 0.0001$; ES = 3.11). FT was significantly lower in DJ30 versus HDJ30 ($Z = 4.845$; $p = 0.0001$, d = 3.79), but significantly higher in DJ40 *vs.* HDJ40 ($Z = 4.437$; $p \leq 0.0001$, d = 3.70) and in DJ50 *vs.* HDJ50 ($Z = 4.549$; $p \leq 0.0001$, d = 4.72).

**Conclusions:** It is concluded that the HDJ requires more time for force production, that the eccentric component requires more time than the concentric and that it is not recommended to use the HDJ over the DJ for reactive purposes. This is the first study that comprehensively compare the differences between DJ and HDJ, which will assist coaches and researchers in the design of future training strategies.

## INTRODUCTION

Plyometrics is a training method that aims to improve muscle power (*Wallace et al., 2010*) in dynamic movements. Plyometric exercises have been conceptualized as fast muscle actions involving the stretch-shortening cycle (SSC) (*Bobbert, Huijing & van Ingen Schenau, 1987b*). This SSC, during plyometric actions, has its basis in that the pre-activated muscle is first stretched (eccentric movement), thus causing the tendon-muscle complex to store elastic energy (*Kuitunen et al., 2007*) and then followed by the shortening action (concentric movement) where this energy can be recoiled in the subsequent concentric (push-off) phase (*Schmidtbleicher, 1992*; *Horita et al., 2003*; *Nicol, Avela & Komi, 2006*; *Kuitunen et al., 2007*). *Schmidtbleicher (1992)* also introduced the concept that fast plyometric exercises are applicable when contact time is less than 250 milliseconds. Usually, the plyometric exercises involve an abrupt (very fast) stretching of muscles after the brake of a free fall (*Bobbert, Huijing & van Ingen Schenau, 1987b*; *Verkhoshansky, 2006*; *Nagano, Komura & Fukashiro, 2007*; *Pedley et al., 2017*). Hence, the inclusion of squat jump and countermovement jump exercises in the plyometric classification (*Loturco et al., 2015a, 2015b*) could be questionable due to the lack of SSC and the absence of impact forces during ground contact time (GCT) after a fall. Thus, they may be called 'ballistic exercises' (*Bosco, Komi & Ito, 1981*).

Comparisons between horizontal and vertical jumps have been the subject of several studies (*Nagano, Komura & Fukashiro, 2007*; *Ball & Zanetti, 2012*; *Loturco et al., 2015a, 2015b*). These comparisons could be based on the specificity principle (*Walshe, Wilson & Ettema, 1998*; *Randell et al., 2010*) and the dynamic correspondence principle (*Verkhoshansky & Siff, 2009*; *Fitzpatrick, Cimadoro & Cleather, 2019*). The latter indicates that using vertical forces could improve the performance of motor actions performed in the vertical component. In the same way, applying horizontal forces would enhance the performance of horizontal movements (*Randell et al., 2010*). Other comparisons between the drop jump (DJ) and the horizontal drop jump (HDJ) showed that the latter is a more complex movement requiring the athlete to consider the optimal angle of projection to achieve the greatest horizontal distance (*Wakai & Linthorne, 2005*). This has been demonstrated by the differences observed in the position and trajectory of the center of mass during the take-off and flight phases in horizontal and vertical jumps (*Nagano, Komura & Fukashiro, 2007*). Additional observations support the notion that vertical plyometric training has a greater effect on the vertical jump test. In comparison, horizontal plyometric training has a greater impact on a horizontal jump test (*Loturco et al., 2015b*). Although these results might be expected, considering the dynamic correspondence principle (*Verkhoshansky & Siff, 2009*), exploring the DJ and HDJ to demonstrate this positioning might also have been appropriate. *Ball & Zanetti*'s *(2012)* study investigated the relationship between the horizontal and vertical reactive strength index, reporting good ICC reliability (r > 0.789) for the DJ *vs.* HDJ. These researchers also reported that GCT from a fall height of 0.4 m is significantly shorter in DJ when compared with HDJ. They also noted that HDJ might be better used to train movements where GCT are longer and

more concentric, such as the acceleration phases of a sprint. Despite the evidence collected in the literature, none of these studies directly compared the DJ against the HDJ.

In this regard, sound reasoning/justification is needed on the different biomechanical magnitudes between DJ and HDJ. This information could help coaches and sports scientists know which jumping exercise involves more or less eccentric and concentric contraction types. Additionally, coaches can use this information to plan their training cycles more effectively by selecting jumping exercises in isolation or combination. In the same line, it has been reported that DJ does not show significant differences in GCT and flight time (FT) with the increase in the fall height (*Bobbert, Huijing & van Ingen Schenau, 1987b*; *Walsh et al., 2004*; *Kipp et al., 2018*), but it is unknown if this behavior is the same for HDJ. In addition, it is also unknown whether the time to peak concentric force ($T_{GRF-2}$) is different for DJ *vs.* HDJ. The latter would allow us to know which exercises require more time until concentric peak force. As far as we could review, we did not find any studies in the literature that reported the effect size of these differences. Furthermore, exercises with different output magnitudes (horizontal and vertical) are commonly used during jumping and sprint training in high-performance athletes. Clarifying these differences could help to determine the athlete's adaptations. Therefore, the present study aimed to compare the spatiotemporal variables of DJ *vs.* HDJ in elite jumpers and sprinters.

## MATERIALS AND METHODS

### Study design

Repeated measures experimental design was applied to test the hypothesis that differences exist between DJ and HDJ plyometric exercises, with the independent variable as the exercise mode, and the dependent variables as the GCT, FT, EPT, CPT, and $T_{GRF-2.}$ Previously, the standing long jump (SLJ) was applied to check whether the distances achieved by the HDJ differ from those of the SLJ and whether the fall height (FH) during the HDJ affects the horizontal distance.

### Subjects

Sixteen male athletes that are jumpers and sprinters (mean ± SD; age = 24.31 ± 2.24 years, body mass = 81.11 ± 5.10 kg, height = 1.86 ± 0.06 m, BMI = 23.44 ±2.21 kg m$^{-2}$ and SLJ = 3.05 ± 0.07 m) were recruited during the general preparation period, all belonging to their national team. Subjects had participated in World (9/16) and European or Pan-American (16/16) championships. All had higher experience in performing plyometric exercises (*Montoro-Bombú et al., 2023*) and following previous recommendations, they abstained from plyometric or strength training in the 3 days prior to the assessment (*Ball & Zanetti, 2012*). They had no history of injuries within the 3 months preceding the measurements nor did they report any orthopedic disorders and medical contraindications to avoid plyometric training. All athletes were informed of the risks associated with the measurements and gave written informed consent. The research was conducted following the recommendations of the latest version of the Declaration of Helsinki (October 2013) and the study was approved by the Ethic Committee of the

Faculty of Sport Sciences and Physical Education of the University of Coimbra (code-CE/FCDEF-UC/00802021 6 July 2021).

## Testing procedures

Participants' height, body mass, and age were collected before the assessment session. Height was measured with a stadiometer accurate to 0.1 cm (Bodymeter 206; SECA, Hamburg, Germany). Body mass was assessed with a SECA scale (Hamburg, Germany), and body mass index was calculated according to previous protocols (*Salami et al., 2010*). A warm-up was performed according to the individual requirements of each athlete. The average duration of the warm-up was 50 min and was divided into two parts. The first part was called general warm-up, where subjects performed joint mobility, approximately 5 min of running, and dynamic flexibility work. The second part was a specific warm-up, where the subjects performed the exercises corresponding to their sports specialties. After the warm-up, all subjects had 5 min of recovery.

For the evaluation of the dependent variables, subjects performed two DJ attempts from three different FH. All athletes were familiar with the procedure. Two DJ attempts at 0.3 m (DJ30), two at 0.4 m (DJ40) and two at 0.5 m (DJ50). After an active rest period which included dynamic stretching exercises, 30-m progressive sprint and SLJ assuring complete recovery, they performed two HDJ attempts at 0.3 m (HDJ30), two at 0.4 m (HDJ40) and two at 0.5 m (HDJ50). The best of each jump, at different FH, was considered for data analysis. The DJ was performed with rebounding (*Bobbert, Huijing & van Ingen Schenau, 1987a*) and both DJ and HDJ keeping the same either arms swinging. For the HDJ attempts it was also required that during the eccentric phase, the athletes remained in a vertical displacement, moving towards the horizontal only after the body reached the lowest point of this phase. This requirement allowed the eccentric phase of both exercises to be determined based on the 0 velocity parameters. Jumps that: (i) presented asymmetric contacts, (ii) presented a GCT greater than 250 ms during the DJ (Not HDJ), or (iii) did not present contact with the force platform were eliminated. This resulted in a total of four eliminated jumps. The SLJ was performed with both feet together and trying to maximize the horizontal jump distance. A line was placed on the force platform as an initial reference. The length of the jump was determined using a metallic tape measure. The distance of the best jump to the nearest 1 cm was measured from the line to the point where the heel landed closest to the starting line, as previously reported (*Almuzaini & Fleck, 2008*). During the HDJ, athletes were informed that contact with the force plate should be as close to the line marked for the SLJ as possible. The covered jump distance was measured following the same indications as the SLJ. It was given a rest interval of 1 min between jumps from the same FH and 4 min between the different FH.

## Instrumentation and data processing

The exercises were performed by landing on a force plate (Kistler Model 9260AA6; Kistler, Winterthur, Switzerland) with a dimension of 0.6 m × 0.4 m × 0.05 m, which was leveled with a custom-made wooden platform (Fig. 1; 4.20 m long, 1.10 m wide, and 0.05 m high). The force plate was configured to collect at a sampling rate of 1,000 Hz using an interface

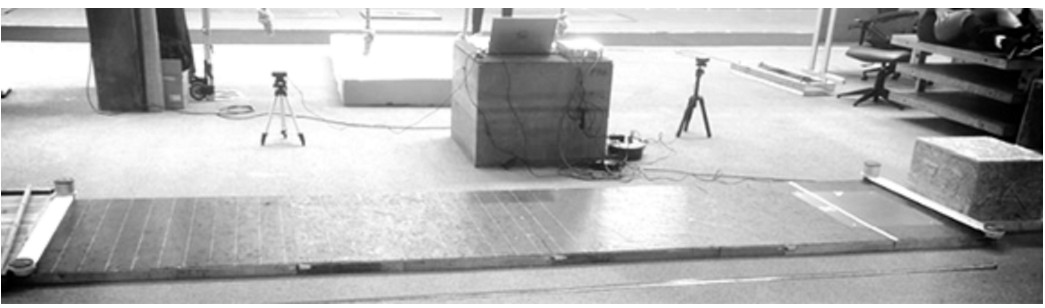

**Figure 1  Wooden platform made for this study.**

box (Kistler Model 9260AA6; Kistler, Winterthur, Switzerland). Data was analyzed using Bioware 5.3.2.9 software (Bioware, Winterthur, Switzerland) following the manufacturer's instructions. An Optojump-nexX30 (Bolzano, Italy) optical contact measurement system (OPT) also was configured to collect and display data in real-time at a sampling rate of 1,000 Hz using an interface. This devise was attached to both edges of the wooden platform, which collected the FT during HDJ.

The GCT is the time difference between touchdown and Take-off; touchdown was defined when the GRFv ascended past 10 N whereas take-off was when the GRFv descended past 10 N' (*Simpson et al., 2018*). In the HDJ, FT was provided by the OPT. The EPT and CPT were bounded when the velocity of the center of mass was close to or equal to zero. Considering that both jumps started with a free fall and that a vertical displacement was requested during the eccentric phase. The delimitation of the eccentric phases was determined from the onset of contact to the moment when the velocity changed direction. To do this, we first calculated static acceleration from the force data and information on body mass (Eq. (1)). Then, we calculate acceleration multiplied by the sampling rate introduced in the force platform plus the initial velocity determined by the free fall (Eq. (2)). The concentric phase was delimited from the moment the velocity changed direction to the moment of takeoff. The TGRF-2 was recognized as the time elapsed from contact until the maximum peak force was achieved in the concentric phase. (Fig. 2).

$$a = \frac{f}{m} - 980,665 \tag{1}$$

$$v = a \times SR + vo \tag{2}$$

where v is the velocity of the center of mass, v0 is the initial landing velocity; a is the acceleration of gravity, f is vertical force and SR is the sampling rate.

## Statistical analyses

Descriptive statistics (mean ± SD) were calculated for each variable (GCT, FT, EPT, CPT, and $T_{GRF-2}$). Comparisons between the two jump exercises were organized in correspondence with the vertical and antero-posterior component of the force platform. The analyses were performed in three groups (A, B and C). Group A: comparison between
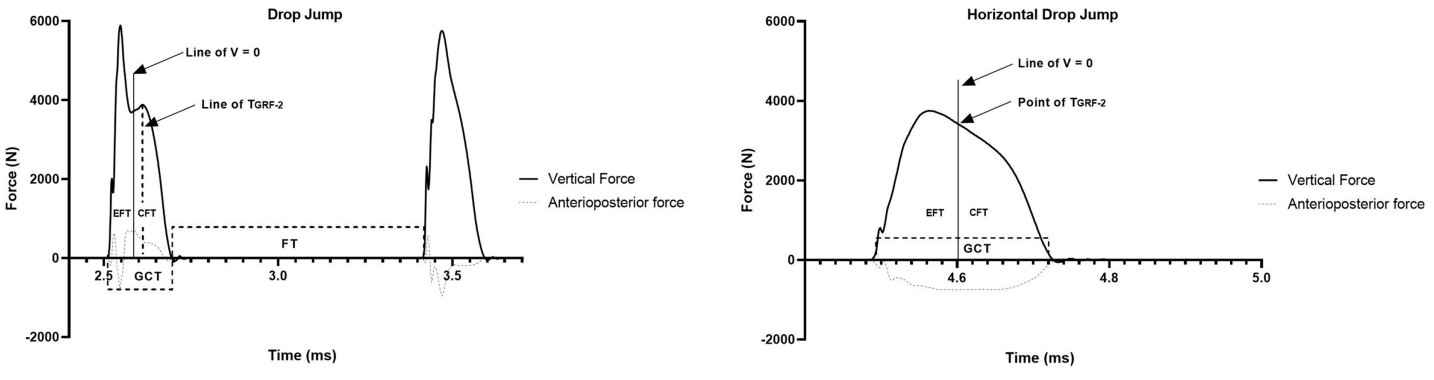

**Figure 2 Identifying for spatiotemporal variables of DJ and HDJ.**     

the vertical component of the DJ relative to (*vs.*) the vertical component of the HDJ (HDJ$_V$) between the same FH and different FH. Group B: comparison between the vertical component of the DJ *vs.* the anteroposterior component of HDJ (HDJ$_a$) between the same FH and different FH. Lastly, group C: comparison between HDJ$_V$ *vs.* HDJ$_a$ between the same FH and different FH. The normality and homogeneity assumptions of the data were not verified. The Wilcoxon test was used to test for statistical differences between DJ *vs.* HDJ for each jump height (pairs and sets). Kruskal Wallis ANOVA was used for within-group comparisons. Values were adjusted using Bonferroni *post hoc* and analyzed to identify statistically significant comparisons set at the α level of $p \le 0.05$. Effect sizes (ES) were analyzed pairwise with G*Power software (v.3.1.9.7; Heinrich-Heine University of Dusseldorf, Germany). Data were analyzed with the statistical package IBM SPSS Statistics (version 27; IBM, Chicago, IL, USA), and graphs were produced with GraphPad (version 9.4.0; GraphPad Software, La Jolla, CA, USA).

## RESULTS

Data are presented by significance level (*p*), ES, and Z-value. Table 1 shows mean data ± SD. GCT analysis (Group A) revealed significantly shorter times ($p \le 0.0149$) in DJ over HDJ and a large ES between the same FH (Box 1), where DJ$_{30}$ *vs.* HDJ$_{30}$ (Z = 3.518; ES = 3.11), DJ$_{40}$ *vs.* HDJ$_{40}$ (Z = 3.519; ES = 1.41) and DJ$_{50}$ *vs.* HDJ$_{50}$ (Z = 3.464; ES = 2.08). ANOVA comparisons for GCT between different FH showed no significant differences in the HDJ ($p \ge 0.05$).

In the FT analysis (Group A), significantly higher FT ($p \le 0.001$) in DJ *vs.* HDJ, the ES also were large for: DJ$_{30}$ *vs.* HDJ$_{30}$ (Z = 3.518; ES = 3.79) in DJ$_{40}$ *vs.* HDJ$_{40}$ (Z = −3.516; ES = 3.70) and in the DJ$_{50}$ *vs.* HDJ$_{50}$ set (Z = 3.517; ES = 4.72), see Box 1. Similarly, significant differences ($p \le 0.001$) were found in all DJ *vs* SLJ sets and large ES in DJ$_{30}$ *vs.* SLJ (Z = 3.517; ES = 3.51), DJ$_{40}$ *vs.* SLJ (Z = 3.517; ES = 3.53) and DJ$_{50}$ *vs.* SLJ (Z= 3.516; ES = 4.28). FT comparisons between different FH for the same plyometric exercise showed no significant differences ($p \ge 0.05$). In addition, the HDJ *vs.* SLJ set also showed no significant differences ($p \ge 0.05$).

During EPT analysis (Group A) significantly shorter values in DJ *vs.* HDJ ($p \le 0.003$) were found in the same FH, with large ES in DJ$_{30}$ *vs.* HDJ$_{30}$ (Z = 3.517; ES = 2.33), in DJ$_{40}$

**Table 1 Mean ± SD of each variable.**

|  | DJ30 | HDJ30$_v$ | DJ40 | HDJ40$_v$ | DJ50 | HDJ50$_v$ | HDJ30$_a$ | HDJ40$_a$ | HDJ50$_a$ |
|---|---|---|---|---|---|---|---|---|---|
| GCT (ms) | 0.206 ± 0.02 | 0.296 ± 0.03 | 0.213 ± 0.03 | 0.263 ± 0.04 | 0.218 ± 0.02 | 0.301 ± 0.05 | - | - | - |
| FT (ms) | 0.703 ± 0.06 | 0.549 ± 0.04 | 0.699 ± 0.04 | 0.547 ± 0.04 | 0.724 ± 0.03 | 0.563 ± 0.03 | - | - | - |
| EPT (ms) | 0.089 ± 0.02 | 0.138 ± 0.02 | 0.085 ± 0.01 | 0.123 ± 0.03 | 0.095 ± 0.02 | 0.143 ± 0.02 | - | - | - |
| CPT (ms) | 0.116 ± 0.01 | 0.157 ± 0.01 | 0.128 ± 0.01 | 0.138 ± 0.01 | 0.123 ± 0.02 | 0.157 ± 0.02 | - | - | - |
| T$_{GRF-2}$ (ms) | 0.109 ± 0.03 | 0.158 ± 0.03 | 0.102 ± 0.02 | 0.134 ± 0.03 | 0.115 ± 0.03 | 0.165 ± 0.06 | 0.201 ± 0.02 | 0.170 ± 0.04 | 0.204 ± 0.07 |

**Note:**
GCT, ground contact time; FT, flight Time; EPT, eccentric phase time; CPT, concentric phase time; T$_{GRF-2}$, time to ground reaction forces 2; DJ30, drop jump of the 30 cm fall height; HDJ30v, horizontal drop jump of the 30 cm fall height only considering the vertical force; HDJ30a, horizontal drop jump of the 30 cm fall height only considering the anteroposterior force; ms, milliseconds.

**Box 1 Summary of the significance level for the differences between drop jump and horizontal drop jump.**

| *→ | Groups | GCT | FT | EPT | CPT | T$_{GRF-2}$ |
|---|---|---|---|---|---|---|
| DJ30 vs. HDJ30$_v$ | A | ↓Y**** | ↑Y**** | ↓Y**** | ↓Y**** | ↓Y**** |
| DJ40 vs. HDJ40$_v$ |  | ↓Y* | ↑Y*** | ↓Y** | ↓Y**** | ↓Y**** |
| DJ50 vs. HDJ50$_v$ |  | ↓Y*** | ↑Y**** | ↓Y** | ↓Y*** | ↓Y**** |
| DJ30 vs. HDJ40$_v$ |  | - | - | ↓Y** | ↓Y*** | ↓Y**** |
| DJ30 vs. HDJ50$_v$ |  | - | - | ↓Y** | ↓Y**** | ↓Y**** |
| DJ40 vs. HDJ50$_v$ |  | - | - | ↓Y*** | ↓Y*** | ↓Y**** |
| DJ30 vs. HDJ30$_a$ | B | - | - | - | - | ↓Y**** |
| DJ40 vs. HDJ40$_a$ |  | - | - | - | - | ↓Y**** |
| DJ50 vs. HDJ50$_a$ |  | - | - | - | - | ↓Y**** |
| DJ30 vs. HDJ40$_a$ |  | - | - | - | - | ↓Y**** |
| DJ30 vs. HDJ50$_a$ |  | - | - | - | - | ↓Y**** |
| DJ40 vs. HDJ50$_a$ |  | - | - | - | - | ↓Y**** |
| HDJ30$_v$ vs. HDJ30$_a$ | C | - | - | - | - | ↓Y**** |
| HDJ40$_v$ vs. HDJ40$_a$ |  | - | - | - | - | ↓Y**** |
| HDJ50$_v$ vs. HDJ50$_a$ |  | - | - | - | - | ↓Y**** |
| HDJ30$_v$ vs. HDJ40$_a$ |  | - | - | - | - | ↓Y**** |
| HDJ30$_v$ vs. HDJ50$_a$ |  | - | - | - | - | ↓Y**** |
| HDJ40$_v$ vs. HDJ50$_a$ |  | - | - | - | - | ↓Y**** |

**Note:**
DJ, drop jump; HDJ$_a$, anteroposterior axis of horizontal DJ; HDJ$_v$, vertical axis of horizontal DJ; GCT, ground contact time; FT, flight Time; EPT, eccentric phase time; CPT, concentric phase time; T$_{GRF-2}$, time to ground reaction forces 2; *, represents the quantity 0 after the point (Ex. Y*** = 0.0007).

vs. HDJ$_{40}$ (Z = 3.517; ES = 1.58) and in DJ$_{50}$ vs. HDJ$_{50}$ (Z = 3.555; ES = 1.53). Significant differences were also found between different FH ($p \leq 0.009$) and large ES were found in DJ$_{30}$ vs. HDJ$_{40}$ (Z = 3.155; ES = 1.33), in DJ$_{30}$ vs. HDJ$_{50}$ (Z= 3.518; ES = 1.74) and in DJ$_{40}$ vs. HDJ$_{50}$ (Z = 3.517; ES = 1.87).

During the CPT analysis (Group A), significantly shorter values in DJ vs HDJ ($p \leq 0.0001$) and large ES were found in DJ vs. HDJ. The DJ30 vs. HDJ30 (Z = 4.664; ES = 2.48), DJ40 vs. HDJ40 (Z = 4.664; ES = 2.54) and DJ50 vs. HDJ50 (Z = 3.816; ES = 1.35). On the other hand, CPT, between the different FH, also found significantly lower differences

**Table 2 Jump distance ± SD (m) with increasing height during the HDJ.**

| SLJ | HDJ$_{30}$ | HDJ$_{40}$ | HDJ$_{50}$ |
|---|---|---|---|
| 3.07 ± 0.05 | 3.10 ± 0.09 | 3.12 ± 0.11 | 3.16 ± 0.07 |

Note:
 SLJ, Standing long jump; HDJ, horizontal drop jump from different heights.

($p \le 0.134$) in DJ over HDJ, with large ES. The DJ30 *vs.* HDJ40 (Z = 3.416; ES = 1.85) DJ30 *vs.* HDJ50 (Z = 4.416; ES = 1.74) and in DJ40 *vs.* HDJ50 (Z = 3.058; ES = 1.87) see Box 1.

The GRF-2 analysis of all groups (A, B and C) showed significant differences, with short durations in DJ *vs.* HDJ and HDJv *vs.* HDJa ($p \le 0.001$). It also showed large ES (between 1.4 and 2.68) with the Z valor approximately equal to 3.518 (Table 1).

In the case of SLJ compared to HDJ at different FH, the distance covered in the jump, showed no significant differences ($p \ge 0.05$). However, it could be observed that there was a very low trend for jump distance to improve with increasing FH (Table 2).

## DISCUSSION

This study aimed to compare the spatiotemporal variables of DJ *vs.* HDJ in elite jumpers and sprinters to test the hypothesis that there are differences between the two exercises. To our knowledge, this is the first comprehensive study to quantify the existing differences of DJ over HDJ in the spatiotemporal variables. The findings show a variety of significant differences between the two exercises primarily focused on group A that compared the vertical component of the DJ *vs.* the vertical component of the HDJ.

Our data showed that GCT (Fig. 3B) was shortest in the DJ when compared with the HDJ, as reported by previous authors (*Ball & Zanetti, 2012*). This could be explained by the greater emphasis on the horizontal projection of the hip during the HDJ, resulting in GCT greater than 250 ms (*Wakai & Linthorne, 2005*; *Nagano, Komura & Fukashiro, 2007*). During DJ, GCT ranges are best in relation to FT found in the literature (*Healy, Kenny & Harrison, 2016*; *Haynes et al., 2019*). As in previous studies (*Bobbert, Huijing & van Ingen Schenau, 1987b*; *Walsh et al., 2004*; *Kipp et al., 2018*), the GCT for the same exercise showed no significant differences between different FH. Our results were encouraging for the coaches, as no athlete saw increased GCT with increasing fall height. This behavior indicates that athletes may not be in the critical zone of plyometric activity, which is proposed as the height at which the athlete shows a significant increase in the GCT related to the performance depending on the stage of preparation. Based on the GCT, our study is consistent with other studies about the importance of using reactivity-based training (DJ with rebounding) with a decrease in GCT which may be favoring speed (*Healy et al., 2019*; *Byrne et al., 2020*) generally used in the special or competitive preparation stage. Furthermore, we highlight that the training based on DJ with countermovement, where greater height is sought in the jump, decreases the reactive component. Also, a previous study (*Bobbert, Huijing & van Ingen Schenau, 1987a*) showed large knee angles, leading to the increase of the GCT. During the HDJ exercise, we found that the GCT are greater than 250 ms. This exercise by their concentric structure do not comply with the principle of reactive jumps (*Verkhoshansky, 2006*; *Flanagan & Comyns, 2008*), but can

accumulate a large eccentric load similar to that required to perform block sets (*Moresi et al., 2011*) of triple and quintuple jumps (*Simpson & Cronin, 2006*; *Holm et al., 2008*). These exercises may be more beneficial as a component of general preparation, where short GCT is not necessary, and training is dominated by general strength and high jump volume. However, they should not be ruled out to be used as accents (reinforcement) during special preparation.

The $T_{GRF-2}$ (Fig. 3E) for the DJ was shorter when compared with a previous study (*Fowler & Lees, 1998*) showing an enhanced performance enabling to reach peak concentric maximal strength. We found that, $T_{GRF-2}$ can take 45–55% of the total jump time, which means that the muscles still must continue producing force after reaching the concentric peak force (45–55% longer). Also, our results showed that in the HDJ the anteroposterior forces take more time to reach the peak force production relative to the vertical forces. This indicates that this could be a good reference if we want to use the HDJ to transfer it to exercises in which the maximum horizontal concentric force is activated after the vertical.

The EPT and CPT results (Figs. 3C and 3D) showed that for both exercises, EPT tends to be significantly lower than CPT. A previous study with elite sprinters also reported this finding (*Coh & Mackala, 2013*). Likewise, EPT is significantly lower in DJ than in HDJ. For CPT, the behavior is the same, where lower significant differences are observed in DJ than in HDJ (Box 1). Another finding is that during HDJ, the GCT of the eccentric component takes longer than the concentric one, significantly affecting the reactive component. Although this understanding is consistent, it is considered that future research should be directed to these issues. Also, when comparing the results between the SLJ pre-test and the distances obtained during the HDJ, it was observed that there were no significant differences between different FH. This result can be explained by the fact that greater fall height increases the levels of neuromuscular pre-activation and the speed reached in the eccentric phase, increasing the contractile potentiation mechanisms of muscles and tendons (*Flanagan & Comyns, 2008*). Likewise, when a muscle undergoes a rapid stretch immediately before a contraction, force amplifies, accompanied by the reuse of the elastic energy stored in the tendon (*Cavagna, 1977*). This behavior could continue to be reproduced as the drop height increases until a critical zone appears in which the time of the eccentric phase is too long for the individual potential. By then, the accumulated elastic energy can be dissipated as heat (*Cavagna, 1977*), which will negatively affect the jump performance by causing a decrease in the distance achieved. This indicates that HDJ could be a good reference to transfer it to exercises in which the maximum horizontal concentric force is activated after the maximum vertical concentric force. Nevertheless, to assure the former action, the takeoff must be performed, flexing the hips and knees in a downward countermovement, rotating the body around the feet to the desired amount of forward lean, and then projecting the body outward and upward by an explosive leg extension as reported above (*Wakai & Linthorne, 2005*). They also noted that in SLJ, the optimal takeoff angle is considerably less than 45°, but during HDJ, these conditions should be better investigated.

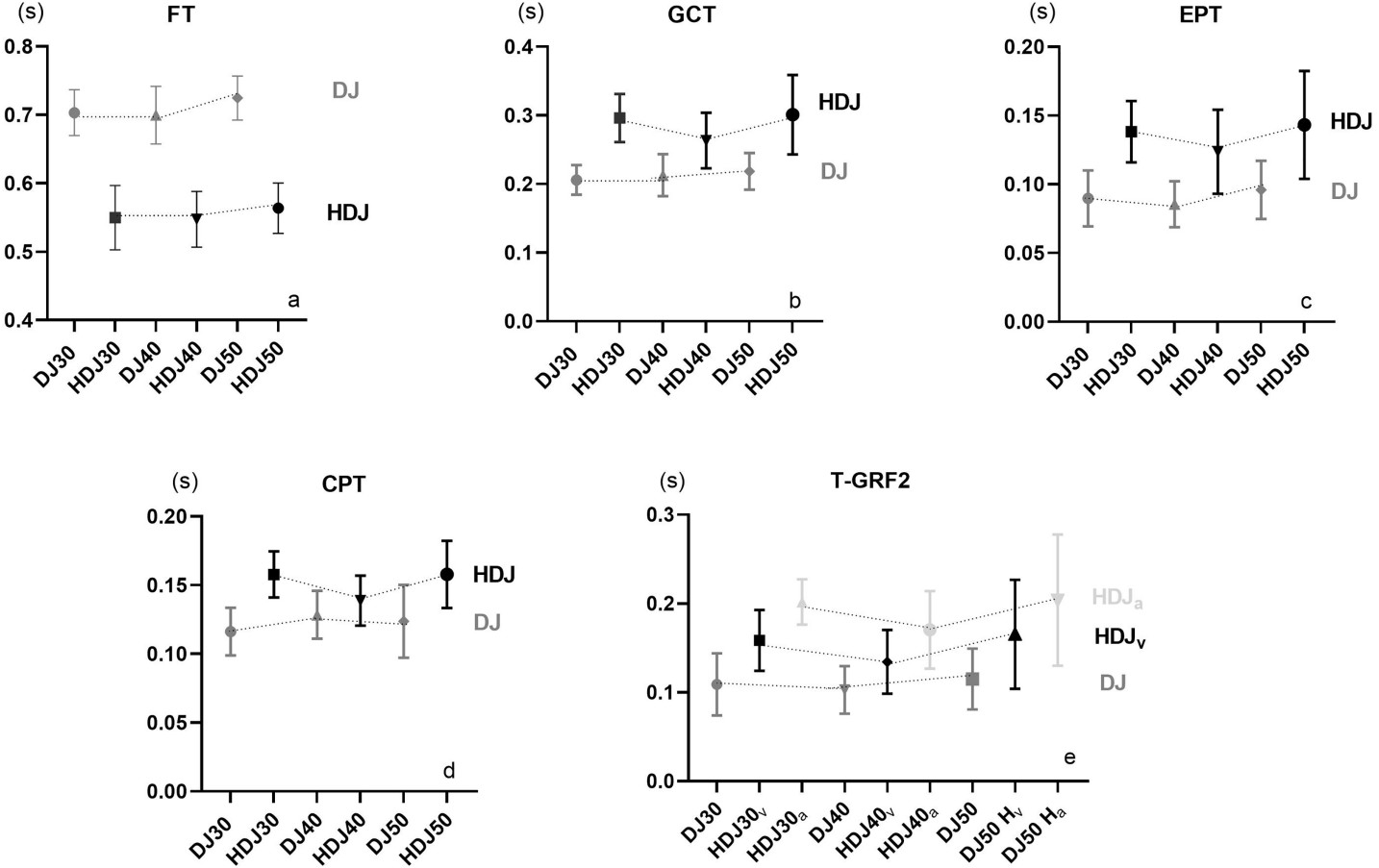

**Figure 3 Graphical representation of the mean ± SD for each variable analyzed.** (A) Mean ± SD of flight time (FT) between the drop jump (DJ) and the horizontal drop jump (HDJ); (B) mean ± SD of ground contact time (GCT) between DJ and HDJ; (C) mean ± SD of eccentric phase time (EPT) between DJ and HDJ; (D) mean ± SD of concentric phase time (CPT) between DJ and HDJ and (E) mean ± SD of time to peak ground reaction force (TGRF-2) between DJ and HDJ.

In addition, FT (Fig. 3A) was found to be significantly higher in the DJ *vs.* the HDJ (Box 1) which supports our hypothesis that there are differences between these exercises. FT was one of the variables that could not be determined with the naked eye, and its behavior differs from that expected by the researchers. Furthermore, as in previous reports (*Bobbert, Huijing & van Ingen Schenau, 1987b*; *Walsh et al., 2004*), no significant differences were found with increasing the FH; on the other hand, the FT of the HDJ was not maintained with a constant mean and therefore reversed after the fall height 0.4 m, possibly due to the increased eccentric load and the need to take more time to move the trunk forward.

Our study is not without its limitations. We recognize that the study could present greater validity if electromyographic delay data and kinematic analysis are added. The lack of randomization of the participant could constitute a limitation; however, it arises from the small number of high-level competitive athletes in these disciplines. These aspects also deserve future studies that will allow coaches to elaborate better and more specific training strategies.

## CONCLUSIONS

This study clearly illustrates the differences between the DJ and the HDJ, with a precise orientation of which exercise should be used according to the season where the athletes are located. As was known, the DJ constitutes a fast stretch-shortening cycle exercise (*Schmidtbleicher, 1992*), but the HDJ can be constituted as a slow stretch-shortening cycle exercise. During the training of maximum velocity, vertical force production and use of the stretch-shorten cycle is going to be of greater precedence to preserve the flight phase; therefore, the DJ should be used. The HDJ has higher contact times and needs more time to attain the peak force. The eccentric component takes longer than the concentric component, being the one that most affect the reactive force parameters. Due to the characteristics of GCT, HDJ might benefit the start and acceleration phase where more TGRF-2 is needed and contact times are longer. The HDJ could not guarantee a faster takeoff time during the support phase in the long jump, but the preparation of quintuple jumps, deca-jumps, and horizontal jumps. This can also be a means of par excellence during the general preparation of a high competition triple jump athlete.

### Funding

The authors received no funding for this work.

### Competing Interests

The authors declare that they have no competing interests.

### Author Contributions

- Raynier Montoro-Bombú conceived and designed the experiments, performed the experiments, analyzed the data, prepared figures and/or tables, authored or reviewed drafts of the article, and approved the final draft.
- Paulo Miranda-Oliveira analyzed the data, authored or reviewed drafts of the article, and approved the final draft.
- Maria João Valamatos performed the experiments, authored or reviewed drafts of the article, and approved the final draft.
- Filipa João performed the experiments, authored or reviewed drafts of the article, and approved the final draft.
- Tom JW Buurke analyzed the data, authored or reviewed drafts of the article, and approved the final draft.
- Amândio Cupido Santos analyzed the data, authored or reviewed drafts of the article, and approved the final draft.
- Luis Rama conceived and designed the experiments, performed the experiments, analyzed the data, prepared figures and/or tables, authored or reviewed drafts of the article, and approved the final draft.

## Human Ethics

The following information was supplied relating to ethical approvals (*i.e.*, approving body and any reference numbers):

The research was conducted following the recommendations of the latest version of the Declaration of Helsinki (October 2013) and the study was approved by the Ethic Committee of the Faculty of Sport Sciences and Physical Education of the University of Coimbra (code-CE/FCDEF-UC/00802021 6 July 2021).

## Data Availability

The raw measurements are available in the Supplemental File.

## Supplemental Information

Supplemental information for this article can be found online at http://dx.doi.org/10.7717/peerj.17026#supplemental-information.

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
