# Peer review of "Spatiotemporal variables comparison between drop jump and horizontal drop jump in elite jumpers and sprinters"

_PeerJ, doi:10.7717/peerj.17026_

## Round 0.1 · original submission · Minor Revisions

Dear Authors

Your submission has been reviewed by two experts in the field of study. The comments of the reviewers are included at the bottom of this letter. We invite you to submit a revised version of the manuscript that addresses the points raised by the reviewers.

We look forward to receiving your revised manuscript.

Best regards

Yung-Sheng Chen, Ph.D.
Academic Editor

**Language Note:** The review process has identified that the English language must be improved. PeerJ can provide language editing services - please contact us at copyediting@peerj.com for pricing (be sure to provide your manuscript number and title). Alternatively, you should make your own arrangements to improve the language quality and provide details in your response letter. – PeerJ Staff

·

Basic reporting

Overall, I very much enjoyed the article. I thought the references were great and used splendidly throughout the article. The tables made sense with your research. Although, I think it would be beneficial for you to go through Box 1 again and either re-label or correct some of the labeling (N=not significant and there was no N found, just -). There were some grammatical errors throughout the paper and I listed them below within each section. I added some suggestions I had for each.

Intro
Line 76 – an instant instead of a instant
Line 78 to 79- "…stretching of muscles after the brake of a free fall…" unclear if you meant break or brace?
Line 101 – change the tenses to all be the same format (compare or compared would be a better fit or expand the sentence)
Line 110 and 111 – investigated the relationship (between) the horizontal...?
Line 117 to 119 – you could condense by saying “but no effect on the opposite” or “but no effect vice versa”
Line 122 – Might try and put that citation somewhere else within the intro
Line 128 to 130 – grammatical error; instead maybe try “the differences could be found in” “how different could the …,…,..., be.”
Line 130 to 132 – grammatical error; …jumping exercises involve more or less amounts of…, or varying amounts; more or less amount needs to be fixed
Line 132 to 133 – reword
Line 142 – could be a determinant or could determine athlete’s adaptations
Overall intro: you use furthermore and therefore a LOT throughout the last paragraph in the intro, try others or spread it out more

Study Design
Line 152 – and the dependent variables as the

Subjects
Line 160 – sixteen male jumpers and sprinters, or sixteen male athletes that are jumpers and sprinters - grammatical error

Testing procedures
Line 185 - you previously mentioned the abbreviated version of fall heights previously in the study design portion line 154
190 and 191 – both DJ and HDJ can keep the same either arms swinging or arm swings for correct grammar

Instrumentation and data processing
Line 210 – data was analyzed…, use was instead of were

Statistical analyses
Line 246 to 248 – either use a comma before group C or change to separate sentences; too long; could do “… FH. Lastly, Group C…”

Results
Line 265 – “…GTC…”, just a typo as I believe you meant GCT
Line 285 – “…showed significantly differences” change to significant
Line 286 – HDJ not DHJ

Discussion
Line 325 – “During the HDJ,…by their…” this sentence needs clarity
Line 382 – “… valuable important…” change to importance

Conclusion
Line 394 – reword or add some more info; “…characteristics do not allow to use it for different purposes.”, add something in between allow and to use or reword (do not allow for multipurpose use maybe)

Experimental design

I think this part was great. If anything, you could maybe just add more justification behind the thought process of why you choose to conduct the experiment the way you did. For instance, why did you choose to have them abstain from plyo exercise for three days? Why not longer or shorter? Also, you talk in testing procedures about jumps being eliminated. How many jumps total were eliminated?

Validity of the findings

No comment

Additional comments

Overall, I thought the research was well done and well described. Just some more clarification in the methodology/figures and some grammatical changes and I think it'll be well done. Congratulations on completing your research and I look forward to seeing more!

·

Basic reporting

This article proposes to investigate drop jump and horizontal drop jump exercises in high level athletes. The paper seems fairly polished, possibly due to it being revised once (I am reviewing revision 1). The rationale is sound.

My main concern with the article is the Introduction lacks focus. The first two paragraphs need to be shortened considerably, combined, and should be just background information about plyometrics. The third and fourth paragraphs can be combined and shortened a little. The best written paragraph, by far, is the final one.

The Procedures in the Methods also need to be separated into more than one paragraph. I suggest starting a new paragraph line 184 with "For the evaluation of the dependent variables..."


The Figures and Tables are appropriate and sufficient.

Experimental design

The study falls within the scope of the journal. The research design is mostly sound.

Preferably the testing order would have been randomized, but considering the timing of the trials I don't think this is disqualifying. This lack of randomization, however, need to be added as a limitation in the Discussion section.

Validity of the findings

The findings appear valid and reported in detail. Statistics used to evaluate the data are appropriate.

Additional comments

Abstract:
line 40: Take out "Anecdotal... that"

Methods:
line 190: You say the drop jump was performed "with rebounding." This would be termed a depth jump, not a drop jump (depth jump includes rebound, drop jump does not). Please correct this terminology throughout the manuscript.

Discussion:
line 382: You can take out the sentence about analyzing more variables. Some studies, such as Wallace et al 2010 that you cite early in the paper, did this for similar exercises.

---

## Round 0.2 · Minor Revisions

Dear Authors

Your submission has been reviewed by two experts in the field of study. The comments of the reviewers are included at the bottom of this letter. We invite you to submit a revised version of the manuscript addressing the points the reviewers raised. Please be aware of that the terminology of depth jump is suggested to use in your study.

We look forward to receiving your revised manuscript.

Best regards

Yung-Sheng Chen, Ph.D.
Academic Editor

·

Basic reporting

I thought you did well and grammatically everything was modified well compared to where it was previously.

Experimental design

No comment.

Validity of the findings

No comment.

Additional comments

Overall, I liked the modifications you made to your manuscript and thought they helped. I think my biggest issue, or question mark, with the manuscript, is the use of the term drop jump instead of depth jump. I understand what you are trying to do with the study and how it was conducted, which is why I think depth jump would be more appropriate.

·

Basic reporting

Thanks to the authors for making many improvements to their manuscript. It reads much better now and some issues have been corrected. There are a few minor things that still need correcting before it is suitable for publication.

-While the authors did make a case for keeping their terminology as "drop jump", it should still be changed to "depth jump." It is much more common in practice, and in recent articles, to use depth jump for the type of landings the authors did in this study. Also, in the rebuttal the authors described that "Drop jumps are executed from lower heights..." depth drops involve a stiff landing with little to no rebound, which is not what the authors did in their study here.

-The authors cite books in their paper and reference list. While Supertaining is a great book for the practitioner, are there any published refereed studies you can cite to make your point in these instances? Same goes for the Verkhoshansky 2006 book cited.


Introduction:
While much better, the Introduction still lacks focus in some areas.
-line 106: Starting with "Cappa and Behm" should be deleted as nothing about your study or what you are talking about in that section has to do with falt vs forefoot landing technique. The sentences beginning "Additional observations support..." and "In comparison, horizontal..." can stay. The rest of the paragraph should be omitted.

line 147: The sentences "Since these are very..." and "The following actions..." don't read well and don't add to your overall point. Please delete.

line 150: Delete "In this sense," and add "... Add "biomechanical" between "of" and "differences" later in the sentence (delete "these") and add "between DJ and HDJ."

line 151: Delete the sentence starting "How different..."

Combine the paragraph beginning "Furthermore, exercises..." with the above paragraph.

Discussion:

-line 405: The sentence beginning "Here, we analyzed..." You should delete this sentence. Other authors have investigated force variables with these exercises. The question of momentum is more appropriate when looking at different heights/distances vs different types of jumps, and even then isn't an important consideration for the practitioner in prescribing training.
-line 407: With the above sentence deleted, the beginning portion of "Despite having..." needs to be deleted as well and the rest of it modified so this sentence just talks about adding EMG as something that should be studied in the future between jump types.

Experimental design

N/A

Validity of the findings

N/A

---

## Round 0.3 · Minor Revisions

Dear Authors

Your submission has been reviewed by two experts in the field of study. Only, one last comment is raised by the reviewers. According to the jump performance described in your study, the terminology of “depth jump” should be used throughout the manuscript. We invite you to submit a revised version of the manuscript addressing the points the reviewers raised.

We look forward to receiving your revised manuscript.

Best regards

Yung-Sheng Chen, Ph.D.
Academic Editor

·

Basic reporting

Thanks to the reviewers for improving their paper.

The response for not changing terminology to "depth jump" instead of "drop jump" is still inadequate and the term should be changed throughout the paper. Until that is done, I will continue to not recommend publication. "Drop jump" is simply not the most common and recognizable way to describe the type of jumps done in this study.

Experimental design

N/A

Validity of the findings

N/A

Additional comments

N/A

---

## Round 0.4 · accepted · Accept

Dear Authors

I would like to express my thanks for your patience and efforts to improve the quality of the manuscript. Although the reviewers point to the use of the terminology of “drop jump”, your response stands in the position to use the consistency of academic works. Your submission is now endorsed for acceptance of publication in PeerJ. Congratulation!!!

Best Regards
Yung-Sheng Chen, Ph.D.
Academic Editor